# Creatine Supplementation in Women’s Health: A Lifespan Perspective

**DOI:** 10.3390/nu13030877

**Published:** 2021-03-08

**Authors:** Abbie E Smith-Ryan, Hannah E Cabre, Joan M Eckerson, Darren G Candow

**Affiliations:** 1Applied Physiology Laboratory, Department of Exercise and Sport Science, University of North Carolina at Chapel Hill, Chapel Hill, NC 27713, USA; saylor16@live.unc.edu; 2Human Movement Science Curriculum, Department of Allied Health Science, University of North Carolina at Chapel Hill, Chapel Hill, NC 27713, USA; 3Department of Exercise Science and Pre-Health Professions, Creighton University, Omaha, NE 68178, USA; joaneckerson@creighton.edu; 4Aging Muscle & Bone Laboratory, Faculty of Kinesiology & Healthy Studies, University of Regina, Regina, SK S4S 0A2, Canada; darren.candow@uregina.ca

**Keywords:** female, dietary supplement, menstrual cycle, hormones, exercise performance, menopause, pregnancy, mood, cognition

## Abstract

Despite extensive research on creatine, evidence for use among females is understudied. Creatine characteristics vary between males and females, with females exhibiting 70–80% lower endogenous creatine stores compared to males. Understanding creatine metabolism pre- and post-menopause yields important implications for creatine supplementation for performance and health among females. Due to the hormone-related changes to creatine kinetics and phosphocreatine resynthesis, supplementation may be particularly important during menses, pregnancy, post-partum, during and post-menopause. Creatine supplementation among pre-menopausal females appears to be effective for improving strength and exercise performance. Post-menopausal females may also experience benefits in skeletal muscle size and function when consuming high doses of creatine (0.3 g·kg^−1^·d^−1^); and favorable effects on bone when combined with resistance training. Pre-clinical and clinical evidence indicates positive effects from creatine supplementation on mood and cognition, possibly by restoring brain energy levels and homeostasis. Creatine supplementation may be even more effective for females by supporting a pro-energetic environment in the brain. The purpose of this review was to highlight the use of creatine in females across the lifespan with particular emphasis on performance, body composition, mood, and dosing strategies.

## 1. Introduction

Dietary supplement use has repeatedly been reported to be highest among educated women, and also appears to increase with age [1]. Creatine has been reported as one of the most commonly used dietary sports supplements. The ergogenic potential of creatine can be attributed to several mechanisms, and may have different effects on males and females. Innately, creatine is an essential substrate for the creatine kinase reaction to catalyze adenosine triphosphate (ATP) production from creatine and phosphocreatine (PCr). This recycling also serves as an endogenous metabolic buffer helping to maintain pH [2], and both mechanisms can support cross-bridge recycling and energy availability during exercise. Creatine concentrations in the central nervous system are also notable, which may support neural function in adaptations to exercise.

Despite widespread use and decades of research related to creatine, its effects in females are not well understood. Creatine characteristics vary between males and females. For example, females have/exhibit 70–80% lower endogenous creatine stores than males [3]. Females have also been reported to consume significantly lower amounts of dietary creatine compared to males [3], indicating that females may benefit from creatine supplementation as a strategy/means to increase endogenous stores. Interestingly, females have higher reported (~10%) resting levels of intramuscular creatine concentrations compared to males [4], which could theoretically lower their responsiveness to supplementation and require higher dosages compared to males [5]. In addition, creatine supplementation has not been shown to effectively reduce amino acid oxidation and measures of protein breakdown following exercise in females, which has been reported in males [6]. Therefore, the ergogenic potential of creatine among females has been questioned.

The current body of literature that has evaluated the effect of creatine supplementation in females suggests that the risk-to-benefit ratio is low [7], with most studies indicating that there are numerous metabolic, hormonal, and neurological benefits. The lack of discussion and exploration of creatine use among females across the lifespan is a disadvantage and missed opportunity, since a better understanding of creatine metabolism pre- and post-menopause yields important implications for improving health and exercise performance for females across their lifetime. Therefore, the aim of this review was to highlight the use of creatine in females from young adulthood to old age.

## 2. Creatine Homeostasis across the Lifespan

As a result of hormone-driven changes throughout various stages of female reproduction, endogenous creatine synthesis, creatine transport, creatine kinase kinetics, and creatine bioavailability are altered over time, highlighting the potential positive implications for dietary creatine supplementation for females [8]. The implications of hormone-related changes in creatine kinetics have been largely overlooked in performance-based studies [8]. Specifically, creatine supplementation may be of particular importance during menses, pregnancy, post-partum, during and post-menopause. The menstrual cycle may influence creatine homeostasis due to the cyclical nature of sex hormone regulation (Figure 1). Studies conducted in animal models have demonstrated that the expression of arginine-glycine aminotransferase (AGAT), the rate limiting step of creatine synthesis, is influenced by estrogen and testosterone levels [9]. Sex hormones, predominantly estrogen and progesterone, have been shown to effect creatine kinase activities and the expression of key enzymes for the endogenous synthesis of creatine [10].

Serum creatine kinase levels are significantly elevated during menstruation [11] compared to non-menstruating years, with creatine kinase levels decreasing with age and pregnancy. The lowest concentrations of creatine kinase values have been reported during early pregnancy (20 weeks or less), equating to about half the concentration found at peak levels (pre-menarche teenage girls) [11,13]. For a more detailed discussion related to CK in women, see Ellery et al. [8]. It has been indicated in rodent models that creatine kinase activity (and possibly creatine metabolism) synchronously increase and decrease with estrogen levels [14]. During the luteal phase when estrogen levels are at their peak, muscle damage may be reduced after eccentric exercise due to creatine kinase sparing [15]. Implications of creatine supplementation and creatine metabolism with respect to the menstrual and reproductive cycle warrant further exploration. The interplay between creatine metabolism and CK kinetics may be particularly important for females with low estrogen concentrations (follicular phase), amenorrhea, during pregnancy, and with the transition to through menopause.

Additional consideration should be given to the metabolic changes associated with a normal menstrual cycle. Estrogen is considered a master regulator of bioenergetics, with the highest levels occurring during the luteal phase of the cycle (begins just after ovulation and goes through the end of the cycle). Protein catabolism and oxidation has been shown to be elevated during this high estrogen phase (luteal); while carbohydrate storage has been shown to be reduced during the luteal phase [16]. Mechanistic support for creatine supplementation has been reported to involve muscle protein kinetics, growth factors, satellite cells, myogenic transcription factors, glycogen and calcium regulation, oxidative stress and inflammation [17,18]. Given increased protein turnover and challenges with glycogen saturation, creatine supplementation may be even more effective in the high estrogen/luteal phase.

## 3. Creatine Use among Pre-Menopausal Women

A considerable amount of evidence indicates that creatine is an effective ergogenic aid for increasing strength, power, and athletic performance in females without marked changes in body weight [5,19,20,21]. The reluctance among females to use creatine may be due to a fear of weight gain or other adverse side effects, which are largely unfounded, particularly in women [20]. This rapid weight gain is more prevalent among males; weight may rapidly and temporarily increase with a loading dose which reflects an increase in cellular hydration (i.e., water weight) [22]. This is a positive aspect for increasing hydration [22]. Weight gain may also result if creatine is consumed with a commonly recommended 1.0 g·kg^−1^ body weight of carbohydrate [17]; this is likely not the best strategy for supplementation in females (see dosing section). When reviewing the literature that has examined the effect of creatine supplementation on a variety of performance indices in females, the benefits firmly outweigh any associated risks or reported adverse events.

The potential for adverse effects from creatine supplementation are largely unfounded. An extensive recent systematic review clearly outlined the lack of adverse effect of creatine supplementation on the gastrointestinal, renal, hepatic, or cardiovascular systems among women supplementing with creatine [20]. The findings in women appear to be similar for men, supporting creatine as a safe, low risk dietary supplement when consumed in recommended doses and regimens [7,20].

Creatine supplementation is most effective for high-intensity, short duration activities or repeated bouts of high-intensity exercise with short rest periods such as jumping, sprinting, and resistance training, since increased levels PCr can more rapidly re-phosphorylate adenosine diphosphate to ATP via the creatine kinase reaction. In addition, PCr buffers hydrogen (H^+^) ions that accumulate during high-intensity exercise and may delay the onset of fatigue. In practice, the increase in intramuscular PCr stores through creatine supplementation allows for a greater stimulus for training which results in physiological adaptations that lead to increases in muscle mass, strength, and muscle fiber hypertrophy [23]. To illustrate the effects of creatine monohydrate (CrM) supplement on exercise performance in females, its relative effects (RE) were calculated and are presented in Figure 2, Figure 3 and Figure 4; RE was calculated using the following equation [24,25]:$$RE=\left(\frac{\left(\frac{{\mathrm{Post}}_{\mathrm{Cr}}}{{\mathrm{Pre}}_{\mathrm{Cr}}}\right)\times 100}{\left(\frac{{\mathrm{Post}}_{\mathrm{PL}}}{{\mathrm{Pre}}_{\mathrm{PL}}}\right)\times 100}\right)\times 100$$
where Pre_PL_ is the pre-test value in the placebo group, Post_PL_ is the post-test value in the placebo group, Pre_Cr_ is the pre-test value in the creating group, and Post_Cr_ is the post-test value in the creatine group. A relative effect greater than 100 represents an increase or improvement in performance with creatine supplementation.

### 3.1. Creatine Supplementation and Strength Performance

Although studies using physically active and highly trained females as participants are lacking, both short- and long-term creatine supplementation has been shown to result in significant improvements in muscular strength and power. One of the first comprehensive studies to investigate the effects of creatine on strength performance in females was conducted by Vandenberghe et al. [26]. Healthy, sedentary females ingested CrM tablets or a maltodextrin placebo (PL) four times daily (20 g∙d^−1^) for 4-d, followed by a maintenance dose (5 g·d^−1^) while participating in a 10-wk(week) resistance training program, three times per wk for 1-hr (5 sets, 12 repetitions @ 70% RM for leg press (LP), shoulder press, squat, leg extension (LE), leg curl, and bench press (BP)), followed by a 10-wk detraining and supplementation phase. The CrM group demonstrated a 6% increase (*p* < 0.05) in muscle PCr following the loading period; and the increase after 5-wk and 10-wk of training was 7% and 10% higher than baseline levels, respectively. There were no significant differences in torque output for the forearm flexors between groups following loading, however, the CrM group demonstrated significantly greater torque values at all other time points (5-wk, 10-wk, detraining) until the end of the study when supplementation had stopped for 4-wk. Results also showed that the increase in 1RM for LP, LE, and squat at 10-wk were 20–25% greater (*p* < 0.05) in the CrM group versus PL. The results for body composition showed that there were no significant differences between groups for BW and % fat; however, the change in fat free mass (FFM) was greater (*p* < 0.05) in the CrM group after both 5-wk (2.0 kg) and 10-wk (2.6 kg) of training compared to PL (1.1 kg and 1.6 kg, respectively). These findings suggest that 10-wk of CrM supplementation was effective for increasing lower-body strength and FFM beyond training alone in untrained females [26], and helped maintain strength during detraining without any significant effects on body weight or percent body fat.

Studies have also shown that CrM is effective for increasing strength performance in trained females. In a study using collegiate female soccer players (*n* = 14), 5-d CrM loading (15 g·d^−1^) followed by a 12-wk maintenance dose (5 g·d^−1^), during 13-wk of resistance training led to significant increases in 1-RM strength for the BP (18%) and squat (24%) compared to PL (9% and 12% increases for BP and squat, respectively) with both groups demonstrating similar increases in BW and FFM [27]. In a related study, Brenner et al. [28] reported significantly greater increases (mean difference: 3.4 kg) in 1-RM BP among NCAA Division I female lacrosse players completing a CrM loading phase (4 × 5 g∙d^−1^ for 7-d (days)), followed by 2 g∙d^−1^ for 4 wks. Both groups demonstrated a similar increase in BW (0.50 kg; *p* < 0.05).

Short-term CrM supplementation has also been shown to significantly increase muscular power in females. In a study by Kambis and Pizzedaz [29], 22 college-aged females randomly received a CrM loading dose relative to FFM (0.5 g∙kg^−1^ FFM divided into four equal doses) or a PL for 5-d and were tested for isokinetic strength of the preferred quadriceps group, thigh circumference, and BW. The results showed that time to peak torque for LE significantly decreased (*p* < 0.05) and that average power in LE and leg flexion significantly increased in the CrM group compared to PL. Similar to the findings of others [26,27,28,30,31], there were no significant differences between groups for changes in body weight, FFM, % fat, as well as mid-quadriceps circumference or skinfold thickness of the measured thigh and suggest that CrM significantly improved muscular power without associated changes in BW or muscle volume.

Not all studies conducted in pre-menopausal women report an additive effect of CrM supplementation on strength performance in females compared to training alone. Ferguson and Syrotuik [30] reported that 10-wk of CrM supplementation (0.3 g·kg^−1^ for 7-d + 0.03 g·kg^−1^ for 9 wk) in combination with resistance training (4 d·wk^−1^ for 9-wk) had no additional effects on strength or body composition compared to training alone in physically active females (18–35 yrs). The results showed that both the PL and CrM groups experienced similar increases in strength and FFM without a significant change in BW over the 10-wk study. The authors suggested that the lack of non-significant findings may have been due to non-responders in the CrM group, an insufficient loading dose, insufficient training volume, or a combination of these factors [30]. Wilborn et al. [31] also recently reported that CrM and whey protein (PRO) supplementation did not enhance training adaptations to an 8-wk split-body strength training program compared to PRO alone in females. The subjects trained 4-d per week and ingested either 24 g PRO (*n* = 9) or 24 g PRO + 5 g Cr (*n* = 8) following each exercise session. The results showed that both groups demonstrated a 2.5 kg increase FFM, with no significant differences between groups for strength or lower body measures of power. The authors [31] acknowledged that the lack of a PL group was a limitation of the study, and suggested that the finding of no additive effect of CrM may have been due to higher baseline levels of intramuscular stores of creatine in their trained subjects, since it has previously been reported that individuals with lower endogenous levels have a greater capacity to increase creatine following supplementation [32].

Although not all studies show that CrM supplementation has an additive effect on adaptations to resistance training compared to training alone, there is substantial evidence to suggest that CrM supplementation is effective for increasing strength and power in both trained and untrained females, without large fluctuations in BW or FFM. Relative effects of creatine supplementation on strength performance demonstrate a consistent improvement in performance compared to placebo (Figure 2). The ergogenic effect of CrM can be attributed to an increase in intramuscular PCr stores that facilitates an increase in training intensity and enhanced recovery between successive bouts of training. A higher training stimulus over time from CrM supplementation results in greater physiological adaptations (i.e., hormonal, increased cell hydration, increased gene expression) that lead to increases in strength and hypertrophy [23].

### 3.2. Effect of Creatine on Exercise Performance

The ergogenic effect of Cr observed in both anerobic and aerobic exercise performance is largely attributed to its ability to prevent fatigue as a result of increased intramuscular stores of PCr, which increases ATP turnover and buffers endogenous H^+^ protons to maintain pH. As a result, a number of studies have also examined the effects of CrM supplementation in females on other anaerobic indices of performance with the majority of studies showing favorable results. For example, creatine loading has been shown to improve anaerobic working capacity (AWC) estimated from the critical power test. AWC represents the maximal work potential associated with the phosphagen energy system (ATP + PCr) and, therefore, provides an estimate of anaerobic power. Eckerson et al. [33] examined the effect of 2-d and 5-d of CrM loading (20 g∙d^−1^) on AWC in physically active females (mean age ± SD = 22 ± 5 yrs) using a double-blind, crossover design and found that 5-d of supplementation resulted in a 22% increase in AWC (*p* < 0.05), whereas the PL trial resulted in a 5% decline in performance. In a follow-up study [34] to determine if phosphate salts had a synergistic effect, AWC was increased by 13.0% and 10.8% following 6-d of loading with CrM or CrM + phosphate salts, respectively, compared to a 1.1% decline in the PL group. These findings were consistent with other studies that used physically active females as participants and reported increases in AWC ranging from 10–15% following CrM loading [35,36] (Figure 3). Tarnopolsky and MacLennan [37] also showed that short-term CrM supplementation (20 g∙d^−1^ × 4-d) increased peak and relative peak anaerobic cycling performance (3.7%) with no gender specific responses in 24 recreationally active males and females. In a related study that used trained participants, Ziegenfuss et al. [23] found that only 3-d of CrM supplementation (0.35 g∙kg^−1^ FFM) increased sprint cycle performance in NCAA Division I athletes and that the effect was greater in females as the sprints were repeated. In a recent study that used amateur soccer players as subjects, Ramirez-Campillo et al. [38] examined the effect of CrM supplementation and 6-wkof plyometric training on jumping, maximal and repeated sprinting, and change of direction speed performance. Females were equally and randomly assigned to one of three groups: CrM + plyometric training; PL + plyometric training; or PL only. The CrM group ingested 20 g∙d^−1^ for 1-wk in four equal doses followed by a single dose for 5-wk, and subjects in the PL groups received glucose in an identical dosing regimen. There were no changes in performance for the PL subjects; both plyometric training groups showed improvements in each of the performance indices, with the CrM group demonstrating greater improvements in jump and repeated sprint performance tests, indicating that adaptations to plyometric training were enhanced with CrM supplementation. Aoki et al. [39] examined the ergogenic effect of CrM on concurrent exercise performance in 14 females (21 ± 2 yrs) who were randomly assigned to receive either CrM (20 g∙d^−1^ × 5-d followed by 3 g∙d^−1^ × 7-d) or PL. Following the 12-d intervention, there were no differences between groups in running performance or 1RM LP following supplementation; however, there was a significant decline in the number of maximal repetitions performed during the last two sets of the repetition max test in the PL group compared to CrM. It was suggested that an increase in intramuscular stores of PCr experienced by the CrM group may have enhanced their recovery from aerobic exercise and improved resistance exercise performance during the repetition max test. The true effect of creatine on recovery has largely been unexplored. Due to sex-based differences in fatigue resistance and recovery, this would be an important area for future research.

Although creatine has not been widely investigated for its effects on endurance exercise performance, there is some evidence to suggest that it may have some ergogenic benefits. Nelson et al. [40] showed that CrM loading for 7 d resulted in a lower oxygen consumption (VO_2_) at submaximal workloads, and reduced the work performed by the cardiovascular system in a study that examined the effects of CrM on cardiorespiratory responses during a graded exercise test (GXT). The results showed that CrM significantly increased total test time (20.3 ± 4 min to 21.5 ± 3.5 min) compared to PL (17.3 ± 3 min to 17.4 min ± 3 min), and that VO_2_ and heart rate at the end of first five stages of the GXT were significantly lower for CrM versus no change for PL. In addition, the ventilatory threshold (VT) increased significantly from pre- to post-testing for the Cr group (66% to 78% peak VO_2_), whereas the PL group demonstrated no change (70% to 68% peak VO_2_). The authors [40] speculated that the decreases in sub-maximal VO_2_ and heart rate were due to increased stores of PCr in muscle, which may have ultimately delayed mitochondrial respiration and lowered VO_2_. In a related study, Smith et al. [41] examined the effect of CrM loading (20 g∙d^−1^ × 5 d) on aerobic power (VO_2_ max) and critical velocity (CV), which is a theoretical velocity that can be maintained for an extended period of time using only aerobic energy stores, and was calculated in their study by having subjects complete four high speed runs to exhaustion at 90, 100, 105, and 110% of peak velocity. The results showed that CrM loading neither positively or negatively influenced VO_2_ max, CV, time to exhaustion, or BW, since there were no significant differences in any of these parameters between the CrM or PL groups. It has also been shown that an increase in PCr levels following CrM supplementation delays the onset of neuromuscular fatigue (NMF), which is characterized by an increase in the electrical activity of the working muscles over time and reflects the progressive recruitment of additional motor units and/or an increase in the firing frequency of motor units that have already been recruited [42,43]. In two separate studies, Stout et al. [43] and Smith et al. [42] showed that 5-d of CrM loading significantly delayed the onset of NMF during incremental cycling exercise compared to PL using both physically active and highly trained female athletes. Both authors suggested that the delay in NMF was due to an increase in intramuscular levels of PCr levels, which may have resulted in a greater capacity to delay anaerobic glycolysis and, in turn, decreased the accumulation of lactic acid and ammonia in the working muscles and the blood. Previous studies using physically active males as subjects have shown that CrM supplementation during 4-wk of high-intensity interval training (HIIT) significantly improved VT [44] and critical power [45] compared to training alone. In contrast, Forbes et al. [46] recently reported that CrM did not augment improvements in cardiorespiratory fitness, performance, or body composition in recreationally active females following a 4-wk HIIT program. In their study [46], 17 females were randomly assigned to receive either CrM (0.3 g∙kg∙d^−1^ × 5-d followed by 0.1 g∙kg∙d^−1^ for 23-d, *n* = 9) or PL (*n* = 8) and completed three HIIT sessions per week for 4-wk with 48-hr between each exercise session. HIIT improved VO_2_peak (CrM = 10.2%; PL = 8.8%), VT (CrM = 12.7%; PL = 9.9%) and time-trial performance (CrM = −11.5%; PLA = −11.6%) with no significant differences between groups. The authors [46] suggested that the differences between their findings and those of Graef et al. [44] and Kendall et al. [45] could be associated with CrM dose, methods to assess endurance performance, and/or sex-based differences.

The studies described above suggest that females with varying levels of training and fitness may experience improvements in both anaerobic and aerobic exercise performance from both short-term and long-term creatine supplementation. Therefore, it seems likely that the ergogenic effects observed in the laboratory would carry-over to competition and allow athletes who must compete in more than one event or game on the same day, or who must compete on successive days, to recover faster and, in turn, optimize performance and improve their probability of winning. Current research suggests creatine is an effective way to improve sport performance in females (Figure 4). Given the difficulty of designing these types of studies, very few investigations have determined how CrM supplementation may influence win-loss records and the execution of skills during competition. Cox et al. [47] examined the effects of short-term CrM supplementation (4 × 5 g·d^−1^ × 6 d) on performance during a field test that simulated soccer match play using elite female soccer players from the Australian National Team and found that the athletes in the CrM group significantly improved repeated sprint performance and some agility tasks that mimicked soccer play compared to the PL group. Many studies have examined the effect of CrM supplementation on swim performance, since it is a sport that allows investigators to more closely mimic competition. While most studies show that supplementation is ineffective for improving single sprint swim performance [48,49,50] in a few studies that required swimmers to perform repeated sprints, improvements in the time to complete the series [51,52] and increases in work and power output [52] were reported.

## 4. Creatine Considerations during Pregnancy

Increased metabolic demand from growth and development during gestation, particularly from the placenta, has been associated with a reduced creatine pool [53]. Recent human data suggests a dramatic alteration in creatine homeostasis during pregnancy [54] and a reduction in creatine stores during pregnancy have been linked with low birth weight and pre-term birth [54,55].

To date, there is growing evidence in animal models that creatine supplementation during pregnancy enhances/augments neuronal cell uptake of creatine and supports mitochondrial integrity in animal offspring, thereby reducing brain injury induced by intrapartum asphyxia [55,56]. Although there are no human studies to date that have evaluated the effect of CrM supplementation during pregnancy, CrM supplementation could provide a safe, low-cost nutritional strategy for reducing intra- and post-partum complications associated with cellular energy depletion [57]. Further details for the mechanisms and implications for creatine use during pregnancy are described in this special [58].

## 5. Creatine Considerations for Post-Menopausal Women

The menopausal related decrease in estrogen is a main contributing factor to the age-related loss in muscle and bone mass [59] and strength (i.e., dynapenia) [60]. While the mechanisms explaining the link between estrogen levels, muscle mass, and strength remain to be determined, there is evidence to suggest that insufficient estrogen levels are associated with increased inflammation and oxidative stress [59,60] and may contribute to the blunted muscle protein synthetic and satellite cell response to anabolic stimuli (i.e., resistance training).

Creatine supplementation has been shown to act as a possible countermeasure to the menopausal related decrease in muscle, bone, and strength by reducing inflammation, oxidative stress, and serum markers of bone resorption, while also resulting in a concomitant increase in osteoblast cell activity (i.e., bone formation) [61,62]. Muscle integrity has also been upregulated with creatine use, resulting in an increase in satellite cell activity, growth factors (i.e., IGF-1), protein kinases downstream in mammalian target of rapamycin pathway, and myogenic transcription factors (for reviews see [7,18,61]). These effects have recently been explored in females, often combined with resistance training. As a result of the unique cyclical and long-term changes in estrogen across the lifespan, creatine supplementation poses an interesting therapeutic strategy for post-menopausal females.

### 5.1. Supplementation Only

Among post-menopausal females (65 ± 2 yrs), a short-term, high dose creatine loading period (0.3 g·kg^−1^·d^−1^ or ~20 g·d^−1^ for 7 days) augmented whole-body FFM (0.52 ± 0.05 kg), muscle strength (LP: 5.2 ± 1.8 kg, BP: 1.7 ± 0.4 kg), and sit-stand and tandem gait test performance [63]. Similar functional improvements were reported in sit-to-stand performance among post-menopausal females (60–80 years) supplementing with a similar creatine dose (0.3 g·kg^−1^·d^−1^ or ~17 g·d^−1^ for 7 days) [64]. Another related study using a similar dose (20 g·d^−1^ loading for 5 days, followed by 5 g·d^−1^ for 23 weeks) reported no differences between CrM and PL on measures of muscle mass, bone mineral, upper- and lower-body strength, or functionality in post-menopausal females (<60 years). Additionally, low-dose chronic supplementation with creatine (1 g·d^−1^ for 52 weeks) among post-menopausal females (58 ± 5 years) failed to have on effect on FFM, bone density, bone turnover, or muscle function, compared to PL [65]. Increasing the dosage of creatine to 3 g·d^−1^ (1 g dose in the morning, afternoon and evening) for an additional 52 weeks (104 weeks in total) also had no effect on the same muscle and bone measures. Furthermore, CrM had no effect on handgrip strength or the number of falls or fractures experienced among older females [66]. Collectively, it appears that a short-term high dose of creatine may have minor effects on muscle and strength among post-menopausal females.

### 5.2. Combined with Resistance Training

The vast majority of research involving creatine supplementation in post-menopausal females has included resistance training as part of the study design, possibly because muscle contractions (i.e., resistance training) lead to greater intramuscular creatine uptake from supplementation [67], which could augment muscle mass and performance. In post-menopausal females (>60 years), Gualano et al. [68] reported that CrM supplementation (*n* = 15; loading phase of 20 g·d^−1^ for 5 days + maintenance phase of 5 g·d^−1^ for 161 days) during supervised whole-body resistance training (7 exercises; 3 sets of 8–12 repetitions) produced greater gains (relative) in appendicular lean tissue mass and BP strength compared to PL (*n* = 15). However, CrM and resistance training had no greater effect on measures of LP strength, bone mineral density or content or serum markers of bone turnover compared to PL and resistance training. No adverse effects were reported from CrM, PL or the resistance training program. Furthermore, when compared to the creatine alone group, the combination of CrM and resistance training resulted in greater muscle accretion and strength (LP and BP) in post-menopausal females [66]. In the longest study to date, Chilibeck et al. [69] showed that CrM supplementation (0.1 g·kg^−1^·d^−1^ or ~7 g·d^−1^) during supervised whole-body resistance training (17 exercises, 3 sets of 10 repetitions, 3 days per week for 52 weeks) reduced the rate of bone mineral density loss in the hip region and increased femoral shaft sub-periosteal width (indicator or bone strength) and upper-body strength in postmenopausal females. Additional supporting work in post-menopausal females (65 ± 5.0 yrs) showed that CrM supplementation (5 g·d^−1^) during 12 weeks of supervised resistance training (8 exercises, 2 sets of 15 repetitions; 3 days/week) significantly increased FFM, strength (BP, leg extension, elbow flexion), and tasks of functionality (30-second chair stand, arm curl test, lying prone-to-stand test) compared to PL [70]. From a clinical perspective, Neves et al. [71] found a beneficial effect from CrM (loading phase: 20 g·d^−1^ for 7 days + maintenance phase: 5 g·d^−1^ for 79 days) during 12 weeks of lower-limb supervised resistance training on lower-limb muscle accretion and physical performance (timed-stand test) in post-menopausal females (55–65 yrs) with knee osteoarthritis compared to PL. In contrast to these studies showing some favorable effects from CrM and resistance training, Candow et al. [62,72] found no greater effect from CrM supplementation (0.1 g·kg^−1^·d^−1^) during 32 weeks of supervised resistance training (11 exercises, 3 sets of 10 repetitions to volitional fatigue; 3 days per week) on measures of muscle mass, bone mineral, or strength in postmenopausal females (<50 years) compared to PL. Furthermore, Pinto et al. [73] failed to show greater effects from CrM supplementation (5 g·d^−1^) and 12 weeks of supervised whole-body resistance training (3 sets of 13–15 repetitions) on measures of bone mineral and muscle strength. However, CrM did augment FFM more than PL over time.

Collectively, these findings suggest that post-menopausal females may experience increases in muscle mass and function when consuming high-dosage creatine (0.3 g·kg^−1^·d^−1^) for at least 7 consecutive days. Creatine supplementation alone or in combination with resistance training appears to provide no benefits in bone physiology in post-menopausal females. However, when combined with resistance training, the vast majority of research supports the efficacy of CrM supplementation (≥5 g·d^−1^) for improving measures of muscle accretion, strength and tasks of physical performance in post-menopausal females. From a safety perspective, creatine poses no greater adverse effects compared to placebo. Future longer-term randomized PL controlled trials with large sample sizes are needed to fully determine whether creatine, with and without resistance training, can positively influence musculoskeletal parameters in post-menopausal females.

## 6. Depression and Mood

Depression rates are two times higher among females compared to males [74]. The increased prevalence of depression among females has been directly linked with hormonal milestones; major depression rates increase during puberty, during the luteal (high estrogen) phase, following pregnancy, and during perimenopause [75]. Despite the hormonal pattern, evidence suggests that this trajectory is not solely dependent on the amount of estrogen and progesterone, but rather how sensitive the brain is to these hormones [76].

Early research examining the role of dysfunctional creatine metabolism in the neurochemical foundations of depression in adults demonstrated a positive relationship between cerebral spinal fluid levels of creatine and dopamine and serotonin metabolites [77,78]. These data suggest that efficient neurotransmission of metabolites affecting mood depends upon the creatine-PCr system functioning properly. The severity of a depressive episode has been inversely linked to white matter creatine and PCr concentrations within the brain suggesting that there is a relationship between brain creatine metabolism and depression [79]. This pattern has been shown to be beneficial for anti-depressant treatment [80], suggesting dietary creatine supplementation may provide a pro-energetic effect in brain chemistry [81] through efficient regeneration of intracellular high-energy phosphates in females.

Previous research has shown that dietary creatine supplementation can promote cell survival and influence the production and usage of energy in the brain [3,82]. Clinical and pre-clinical evidence has reported positive effects of creatine supplementation on mood by restoring brain energy levels and homeostasis. Altered brain bioenergetics and mitochondrial dysfunction have been linked with depression, particularly as it relates to CK, ATP, and inorganic phosphate (P_i_). In terms of energy usage, in vivo measurements of the female adult brains with a major depressive disorder, demonstrate a distinctive pattern of energy-related metabolites, specifically, a decrease in beta-nucleoside triphosphate and increased PCr level resulting from an increased use of ATP [83]. Supplementation with CrM has also been shown to significantly augment cerebral PCr and P_i_ [84]. Females have been reported to have lower levels of creatine in the brain, particularly the frontal lobe [85], which controls mood, cognition, memory, and emotion. As a result of sex-differences in brain creatine concentrations, supplementation may be even more effective for females for supporting a pro-energetic environment in the brain. When combined with regular antidepressant use, 8-weeks of CrM supplementation reduced depressive symptoms in female adolescents and adults with major depression [86,87,88]. In healthy adolescent females taking anti-depressant medication, the mean Children’s Depression Rating Scale-Revised (CDRS-R) score declined from 69 to 30.6, a 56% decrease, in those consuming 4 g of CrM daily for 8 weeks [81]. Similar results were demonstrated in healthy adult females taking anti-depressant medication. There were significantly greater improvements in Hamilton Depression Rating Scale (HAM-D) score, with improvements observed after 2 weeks in females consuming 5 g∙d^−1^ of creatine for 8 weeks [84]. This response time accelerates the effectiveness of antidepressant medications compared to the typical 4–5 weeks acclimatization to detect/identify effects with these therapeutics alone [88,89]. Dietary creatine intake is inversely proportional with depression occurrence; with a 31% greater incidence of depression in adults in the lowest quartile of creatine intake [90]. Increasing creatine concentrations in the brain as a result of increased animal protein consumption and, more effectively, through CrM supplementation, has strong evidence to support mood and depression, particularly in females. This has important relevance through various stages across the lifespan that demonstrate increased prevalence in depression as a result of cyclical hormones, including puberty, post-partum, and menopause.

## 7. Cognition and Sleep

Brain activity results in a rapid reduction in PCr levels to maintain ATP levels [91]. Therefore, during periods of high mental stress, which require a higher PCr demand, ATP turnover may be impaired. Creatine supplementation has been shown to support greater neural ATP resynthesis, which provides a cognitive advantage for tasks that rely on the frontal cortex (i.e., cognition, attention, memory) [92]. Brain creatine concentrations appear to be variable based/depending upon age, lifestyle choices, diet, and other factors [93], which is relevant when considering creatine supplementation for females across the lifespan.

Creatine supplementation in humans has consistently demonstrated improved cognitive performance and brain function and reduces mental fatigue during stressful mental tasks in healthy adults [94]. Greater cognitive improvements as a result of creatine supplementation have also been reported in individuals with cognitive impairments [95,96]. Creatine supplementation also appears to improve cognitive function in vegetarians due to lower brain creatine concentrations. Females process stress different than males [97], often practicing more frequent habits of multi-tasking [98] and are also more often susceptible to sleep deprivation due to pregnancy, post-partum demands, and menopausal sleep disturbances. Creatine supplementation has been shown to support these exact scenarios by augmenting mental capacity under sleep deprivation [96]. Additionally, acute and chronic sleep deprivation appear to be more detrimental to females compared to males; with lower levels of alertness and increased sleepiness-related risks [99]. Sleep deprivation has been reported to result in lower cognition, as well as a reduction in sleep quality for females during the follicular phase (low estrogen), which is also when creatine kinase levels appear to be lowest [100]. As a result of this promising research, creatine supplementation throughout the menstrual cycle may aid in attenuating its adverse effects on cognition and sleep. Cognitive and sleep benefits of creatine supplementation may be most helpful during periods of high stress and sleep deprivation.

## 8. Dosing Strategies

Supplementing with CrM can be accomplished using two strategies, both resulting in similar increases in intramuscular PCr levels (Table 1) [7]. A loading phase in females results in a 19% increase in total muscle creatine concentrations [67,101], which is similar to the response for males. A daily dose of 5 g is also equally effective for increasing muscle creatine stores, however this approach requires more time (~3–4 weeks) compared to a traditional loading approach (5 days) [2]. When practicing a loading phase, creatine remains elevated for about 30 days following completion of supplementation, which appears to be the same for both males and females. Based on available evidence, it appears that females can practice the same dosing strategy that is recommended for males. With creatine supplementation, there is an individual variability with the response to creatine saturation with responders, quasi-responders, and non-responders [102]. This has not yet been explored in females, but it is assumed that all individuals may respond differently.

Skeletal muscle creatine uptake can also be influenced by insulin availability, which may enhance creatine retention [103,104]. Consuming creatine with carbohydrate (~50 g) and protein (~50 g) [105], or with 1 g∙kg^−1^ of glucose, may increase total muscle creatine concentrations compared to creatine supplementation alone [106]. However, in females, the additional calories from CHO/PRO to enhance ingestion, particularly during a loading phase, may not be warranted. Specifically, for women who tend to burn fewer calories than men, if additional calories are not needed to meet training needs, the benefit from enhanced absorption, does not outweigh the potential enhanced absorptive effect; creatine monohydrate has extremely high bioavailability [107]. Additionally, due to the menstrual cycle, the lower CHO oxidation in the follicular phase may suggest the added macronutrients are not needed. One strategy is to ingest creatine with a usual meal or add it to a protein shake due to the insulin properties of amino acids.

Brain concentrations of creatine and PCr are elevated as a result of 0.13–0.80 g/kg/day for 14 days [84]. To maximize brain uptake, a loading phase (15–20 g∙day^−1^ for 3–7 days) followed by a consistent regular daily 5–10 g dose is optimal for tissue saturation. Peak absorption of creatine from supplementation is optimized when ingested as a solution vs. capsule, lozenge, or solid meat [108].

## 9. Conclusions and Application

As a result of changes in creatine homeostasis across the lifecycle, particularly as it relates to estrogen, creatine supplementation appears to provide many potential benefits for females. Creatine use has consistently demonstrated improvements in muscle and brain PCr levels, which has been shown to result in improvements in strength and exercise capacity. When combined with resistance training, creatine further augments body composition and bone mineral density, particularly in post-menopausal females. Creatine supplementation has also been shown to improve mood and cognition. A traditional loading dose (0.3 g∙day^−1^ 5–7 days) or a routine daily dose (5 g) for 4 weeks can be effective for females. For brain saturation, higher doses (15–20 g∙day^−1^ for 3–7 days, followed by 5–10 g∙day^−1^) of creatine are warranted. Future data evaluating more specific effects of creatine across the menstrual cycle will help to more clearly understand the varied benefits at different phases of the menstrual cycle and potential for use across the lifespan.

## Figures and Tables

**Figure 1 nutrients-13-00877-f001:**
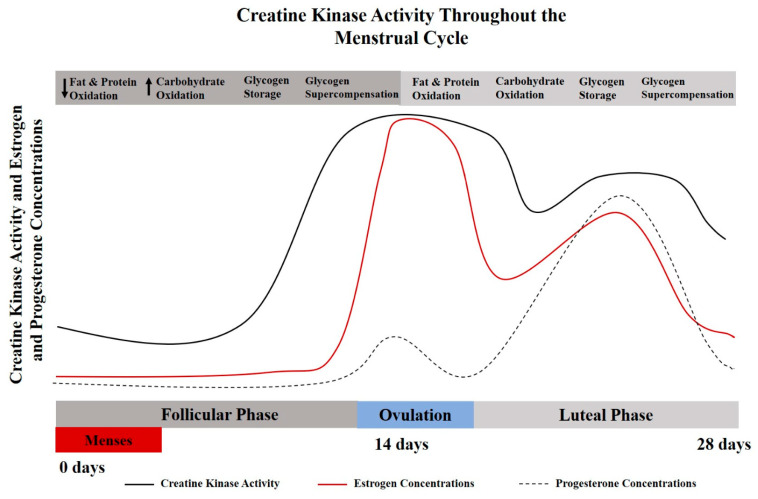
A theoretical model for the interplay between creatine kinase and menstrual cycle hormones [6,10,11]. Creatine metabolism and creatine kinase concentrations vary throughout the menstrual cycle and lifecycle. These alterations may also influence metabolic characteristics of protein and carbohydrate oxidation [12], which provides a physiological basis for the potential use of creatine supplementation for females.

**Figure 2 nutrients-13-00877-f002:**
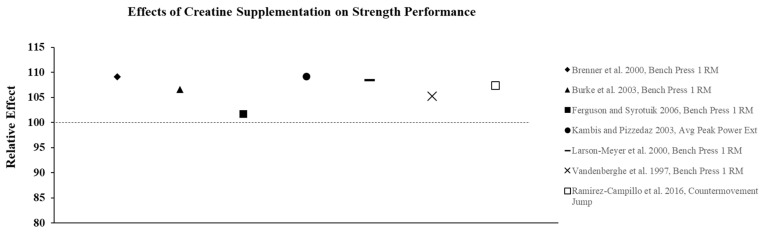
The relative effects of creatine supplementation in comparison to placebo for strength performance in females.

**Figure 3 nutrients-13-00877-f003:**
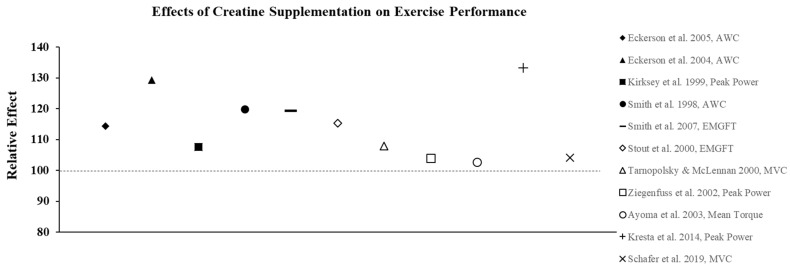
The relative effects of creatine supplementation in comparison to a placebo on exercise performance in females.

**Figure 4 nutrients-13-00877-f004:**
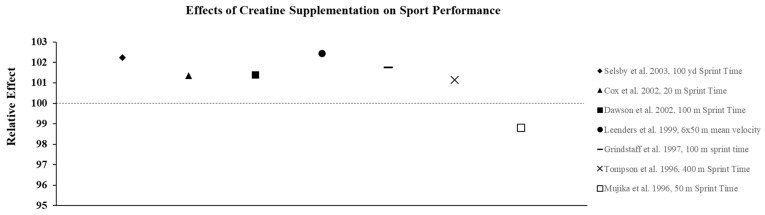
The relative effects of creatine supplementation in comparison to placebo on sports performance in females.

**Table 1 nutrients-13-00877-t001:** Dosing guidelines for creatine supplementation in females [2,67,84,101].

	Dose	Maintenance
Loading Dose	5 g 4 × daily (20 g/day; 0.3 g/kg/day) every 4 h for 5 days	3–5 g (0.03 g/kg/day) daily
Example: 150 lb female (68.2 kg)	8:00 am: 5 g	3–5 g (2.0 g/kg) daily
	12:00 pm: 5 g	
	4:00 pm: 5 g	
	8:00 pm: 5 g	
Routine-Consistent Dose	5 g daily	5 g daily
Brain Saturation	15–20 g daily for 3–7 days (in divided dose)	5–10 g daily

Menstrual Cycle Notes: Due to elevated protein turnover in the luteal phase, creatine supplementation/loading may support muscle protein preservation.

## Data Availability

No new data were created or analyzed in this study. Data sharing is not applicable to this article.

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
