# Peer review of "Creatine Supplementation in Women’s Health: A Lifespan Perspective"

_nutrients, 2021, doi:10.3390/nu13030877_

Round 1

Reviewer 1 Report

The paper describes the effects of creatine supplementation in women in different stages of their reproductive life. The paper is easy to follow and potentially interesting. 
I have some comments for the authors
Line 87 how do you explain peak levels in pre-menarche teenage girls with synchronously high levels with high estrogens levels?
Line 108 and 350 suggest women instead of females
Lines 109-115 provide references
Chapter 3.1: In reported studies was diet assessed? As it can be a relevant interferent. 
Lines 358-361 provide references
I suggest introducing a paragraph for side effects with particular attention to available data in the long term use and to women with impaired kidney function. Also, provide safety data for women with normal kidney function. 
What about the cardiovascular effects of creatine supplementation?

Author Response

Thank you for the opportunity to review and resubmit. We appreciate the constructive comments; all changes have been highlighted throughout the text.

Reviewer 1:

The paper describes the effects of creatine supplementation in women in different stages of their reproductive life. The paper is easy to follow and potentially interesting. 
I have some comments for the authors
Line 87 how do you explain peak levels in pre-menarche teenage girls with synchronously high levels with high estrogens levels?

This is a really important question. The data is mixed regarding the direct relationship between the cyclical nature of estrogen and CK.  The higher levels of CK in premenarche may match the rise in estrogen just prior to menarche.  However, due to the different isoforms of CK, it may be related to another isoform (mitochondrial, brain, or ubiquitous, vs. muscle).  Since this discussion of CK specifically is parallel to the purpose of the paper, we have included an additional reference for the reader to refer to for additional data on CK in women (Ellery et al. 2016).

Line 108 and 350 suggest women instead of females

Thank you. This has been updated.

Lines 109-115 provide references

We appreciate your prompt for the inclusion of references. The following references have been added:

Mihic, S.; MacDonald, J.R.; McKenzie, S.; Tarnopolsky, M.A. Acute creatine loading increases fat-free mass, but does not affect blood pressure, plasma creatinine, or CK activity in men and women. Med Sci Sports Exerc 2000, 32, 291-296, doi:10.1097/00005768-200002000-00007.

Branch, J.D. Effect of creatine supplementation on body composition and performance: a meta-analysis. Int J Sport Nutr Exerc Metab 2003, 13, 198-226.

de Guingand, D.L.; Palmer, K.R.; Snow, R.J.; Davies-Tuck, M.L.; Ellery, S.J. Risk of Adverse Outcomes in Females Taking Oral Creatine Monohydrate: A Systematic Review and Meta-Analysis. Nutrients 2020, 12, doi:10.3390/nu12061780.

Sobolewski, E.J.; Thompson, B.J.; Smith, A.E.; Ryan, E.D. The physiological effects of creatine supplementation on hydration: a review. American Journal of Lifestyle Medicine 2011, 5, 320-327, doi:10.1177/1559827611406071.

Chapter 3.1: In reported studies was diet assessed? As it can be a relevant interferent. 

Diet in some of these studies were evaluated and agree that this is an important confounder; however, the data with creatine suggests despite dietary creatine, creatine supplementation can improve muscle creatine saturation.  To detail this in these studies will distract from the outcomes and is beyond the scope of the present paper.  We think it is best to stand on the science that was presented, and assume that each study was designed appropriately.

Lines 358-361 provide references

References to support the effect of creatine supplementation on bone have been included. The following references have been added:

Candow, D.G.; Forbes, S.C.; Chilibeck, P.D.; Cornish, S.M.; Antonio, J.; Kreider, R.B. Effectiveness of Creatine Supplementation on Aging Muscle and Bone: Focus on Falls Prevention and Inflammation. J Clin Med 2019, 8, doi:10.3390/jcm8040488.

Candow, D.G.; Forbes, S.C.; Vogt, E. Effect of pre-exercise and post-exercise creatine supplementation on bone mineral content and density in healthy aging adults. Experimental gerontology 2019, 119, 89-92, doi:10.1016/j.exger.2019.01.025.

I suggest introducing a paragraph for side effects with particular attention to available data in the long term use and to women with impaired kidney function. Also, provide safety data for women with normal kidney function. 
What about the cardiovascular effects of creatine supplementation?

We sincerely appreciate this comment.  There is a recent systematic review that extensively covers the potential risk for creatine supplementation in women.  This section of the manuscript has been updated to more clearly support the data presented on the lack of adverse events with creatine supplementation in women.  Additionally, a small section has been included to address potential gastrointestinal, hepatic, renal, and cardiovascular effects.

This section now reads:

A considerable amount of evidence indicates that creatine is an effective ergogenic aid for increasing strength, power, and athletic performance in females without marked changes in body weight [5,18,19]. The reluctance among females to use creatine may be due to a fear of weight gain or other adverse side effects, which are largely unfounded, particularly in women [19]. This rapid weight gain is more prevalent among males; weight may rapidly and temporarily increase with a loading dose which reflects an increase in cellular hydration (i.e. water weight) [20]. This is a positive aspect for increasing hydration [20]. Weight gain may also result if creatine is consumed with a commonly recommended 1.0 g·kg−1 body weight of carbohydrate [16]; this is likely not the best strategy for supplementation in females (see dosing section). When reviewing the literature that has examined the effect of creatine supplementation on a variety of performance indices in females, the benefits firmly outweigh any associated risks or reported adverse events.

The potential for adverse effects from creatine supplementation are largely unfounded. An extensive recent systematic review clearly outlined the lack of adverse effect of creatine supplementation on the gastrointestinal, renal, hepatic, or cardiovascular systems among women supplementing with creatine [19]. The findings in women appear to be similar for men, supporting creatine as a safe, low risk dietary supplement when consumed in recommended doses and regimens [7,19].

Reviewer 2 Report

Smith-Ryan et al. provide a narrative review on the use of creatine supplements for women, focusing on strength and exercise performance, depressive disorders, sleep and cognition. Due to established alterations to creatine metabolism driven by sex-hormones (specifically estrogen), the authors discuss the efficacy of creatine supplements for women across the reproductive lifespan (mainly pre and post-menopausal). 

Overall, this review nicely summarises the available literature. I have the following comments, which I hope the authors agree will improve the manuscript slightly before publication. 

  • The title is somewhat misleading. As the review's focus is on creatine supplement use and not endogenous creatine metabolism, this needs to be made clear in the title.
  • Figure one is a little blurry. It would also be useful to provide references to the source data used to generate the theoretical model. 
  • Line 84, I think you need to clarify that you refer to serum/plasma creatine kinase levels. One may assume that you are referring to skeletal muscle or other tissue. 
  • Line 112, there was a recent systematic review and meta-analysis on adverse-effects of creatine supplementation in females (de Guingand et al., 2020). This should be referenced to provide evidence for your statement on safety and lack of weight gain in women taking creatine. 
  • line 133, missing a space between PrePl and is
  • line 151, should read ' the CrM group demonstrated.'
  • line 183, it might be useful to restate that this section is focussed on young pre-menopausal women because apart from the title, this isn't really emphasised, i.e. 'not all studies conducted in premenopausal women report........'
  • line 318, CR not defined. You used CrM or creatine throughout and should keep this language consistent. 
  • Lines 329-332, should they be there? It's not clear what message you are trying to convey, and its a rather abrupt end to the section. 
  • Lines 345-346, the statement seems to be incomplete and thus is hard to follow. Note, the pregnancy review has now been published and can be referenced (Muccini et al., 2021)
  • lines 506-507 do not make sense to be. Should it read cognition instead of cognitive?
  • References would be useful for Table 1, especially the comment about the menstrual cycle. 
  • Line 525, please explain why in females, CHO/PRO ingestion is not as important in females. 
  • In section 8, it is not always clear whether the data you are quoting is from male or female studies or both. Presumably, the point of this section is to directly compare dosing regimens between males and females with a final recommendation of what appears to work best for females? This is not abundantly clear and should be revised. 
  • Finally, there a quite a few typographical errors (missing spaces, double spacing, changes in font style) throughout. Please work closely with the editorial team to fix these issues before publication, as they are somewhat distracting while reading through the manuscript. 

Author Response

Reviewer 2

Smith-Ryan et al. provide a narrative review on the use of creatine supplements for women, focusing on strength and exercise performance, depressive disorders, sleep and cognition. Due to established alterations to creatine metabolism driven by sex-hormones (specifically estrogen), the authors discuss the efficacy of creatine supplements for women across the reproductive lifespan (mainly pre and post-menopausal). 

Overall, this review nicely summarises the available literature. I have the following comments, which I hope the authors agree will improve the manuscript slightly before publication. 

We sincerely appreciate the reviewer’s time toward making this a better, more effective paper.  Each suggestion has been addressed with changes highlighted within the updated manuscript. 

The title is somewhat misleading. As the review's focus is on creatine supplement use and not endogenous creatine metabolism, this needs to be made clear in the title.

We agree; the title has been updated to read: Creatine Supplementation in Women’s Health: A Lifespan Perspective

Figure one is a little blurry. It would also be useful to provide references to the source data used to generate the theoretical model. 

Great idea.  References have been added.  We have also tried to update the figure for readability.  We will work with the editorial team to integrate a more legible figure.

Line 84, I think you need to clarify that you refer to serum/plasma creatine kinase levels. One may assume that you are referring to skeletal muscle or other tissue. 

These statements have been updated to clarify the data being discussed related to serum/plasma creatine kinase levels.

Line 112, there was a recent systematic review and meta-analysis on adverse-effects of creatine supplementation in females (de Guingand et al., 2020). This should be referenced to provide evidence for your statement on safety and lack of weight gain in women taking creatine. 

This is an important suggestion and was inadvertently left out on our part. This reference, and a few others, have been included to further explain the work that has been done, and the impact of creatine supplementation on adverse effects in women.

line 133, missing a space between PrePl and is

The space has been included.

line 151, should read ' the CrM group demonstrated.'

‘group’ has been added after CrM

line 183, it might be useful to restate that this section is focussed on young pre-menopausal women because apart from the title, this isn't really emphasised, i.e. 'not all studies conducted in premenopausal women report........'

This is a great suggestion.  This has been updated per your suggestion to more clearly note these studies are focused on pre-menopausal women.

line 318, CR not defined. You used CrM or creatine throughout and should keep this language consistent. 

Thank you for this catch.  CR has been changed to CrM

Lines 329-332, should they be there? It's not clear what message you are trying to convey, and its a rather abrupt end to the section. 

We apologize for this oversight.  This text was maintained from the transfer of our manuscript into the Nutrients template.  This extraneous text has been removed.

Lines 345-346, the statement seems to be incomplete and thus is hard to follow. Note, the pregnancy review has now been published and can be referenced (Muccini et al., 2021)

This section has been updated and the Muccini reference has been added.  These lines now read: Although there are no human studies to date that have evaluated the effect of CrM supplementation during pregnancy, CrM supplementation could provide a safe, low-cost nutritional strategy for reducing intra- and post-partum complications associated with cellular energy depletion [55].

lines 506-507 do not make sense to be. Should it read cognition instead of cognitive?

These lines now read:  Sleep deprivation has been reported to result in lower cognition, as well as a reduction in sleep quality for females during the follicular phase (low estrogen), which is also when creatine kinase levels appear to be lowest [98].

References would be useful for Table 1, especially the comment about the menstrual cycle. 

References have been included. Thank you for this suggestion.

Line 525, please explain why in females, CHO/PRO ingestion is not as important in females. 

This section has been updated to address this comment.  The following information has been included: Specifically, for women who tend to burn fewer calories than men, if additional calories are not needed to meet training needs, the benefit from enhanced absorption, does not outweigh the potential enhanced absorptive effect; creatine monohydrate has extremely high bioavailability [105]. Additionally, due to the menstrual cycle, the lower CHO oxidation in the follicular phase may suggest the added macronutrients are not needed.  

In section 8, it is not always clear whether the data you are quoting is from male or female studies or both. Presumably, the point of this section is to directly compare dosing regimens between males and females with a final recommendation of what appears to work best for females? This is not abundantly clear and should be revised. 

Thank you for this comment.  The foundational work with creatine saturation was done on a mixed sample of men and women.  This has made more clear that the results have also been conducted in females. Based on the data, both men and women can effectively supplement with the same regiment. This section has been updated.

Finally, there a quite a few typographical errors (missing spaces, double spacing, changes in font style) throughout. Please work closely with the editorial team to fix these issues before publication, as they are somewhat distracting while reading through the manuscript. 

We really appreciate this comment, and apologize for the distracting errors.  When transferring the manuscript to the Nutrients specific template, it appears there were several areas that caused some changes to spacing and fonts.  We apologize for this.  We have reviewed this several times to try to account for these formatting errors.

Round 2

Reviewer 1 Report

Thank you for the revision. I am okay with the changes made according to the reviewers' comments.